

# Multi-dimensional Precision Livestock Farming: a potential toolbox for sustainable rangeland management

Agustina di Virgilio[1,2], Juan M. Morales[1], Sergio A. Lambertucci[2], Emily L.C. Shepard[3] and Rory P. Wilson[3]

[1] Grupo de Ecología Cuantitativa, INIBIOMA, CONICET-UNCO, Bariloche, Río Negro, Argentina
[2] Grupo de Investigaciones en Biología de la Conservación, INIBIOMA, CONICET-UNCO, Bariloche, Río Negro, Argentina
[3] Department of Biosciences, University of Wales, Swansea, United Kingdom

Corresponding author
Agustina di Virgilio,
adivirgilio@comahue-conicet.gob.ar

## ABSTRACT

**Background**. Precision Livestock Farming (PLF) is a promising approach to minimize the conflicts between socio-economic activities and landscape conservation. However, its application on extensive systems of livestock production can be challenging. The main difficulties arise because animals graze on large natural pastures where they are exposed to competition with wild herbivores for heterogeneous and scarce resources, predation risk, adverse weather, and complex topography. Considering that the 91% of the world's surface devoted to livestock production is composed of extensive systems (i.e., rangelands), our general aim was to develop a PLF methodology that quantifies: (i) detailed behavioural patterns, (ii) feeding rate, and (iii) costs associated with different behaviours and landscape traits.

**Methods**. For this, we used Merino sheep in Patagonian rangelands as a case study. We combined data from an animal-attached multi-sensor tag (tri-axial acceleration, tri-axial magnetometry, temperature sensor and Global Positioning System) with landscape layers from a Geographical Information System to acquire data. Then, we used high accuracy decision trees, dead reckoning methods and spatial data processing techniques to show how this combination of tools could be used to assess energy balance, predation risk and competition experienced by livestock through time and space.

**Results**. The combination of methods proposed here are a useful tool to assess livestock behaviour and the different factors that influence extensive livestock production, such as topography, environmental temperature, predation risk and competition for heterogeneous resources. We were able to quantify feeding rate continuously through time and space with high accuracy and show how it could be used to estimate animal production and the intensity of grazing on the landscape. We also assessed the effects of resource heterogeneity (inferred through search times), and the potential costs associated with predation risk, competition, thermoregulation and movement on complex topography.

**Discussion**. The quantification of feeding rate and behavioural costs provided by our approach could be used to estimate energy balance and to predict individual growth, survival and reproduction. Finally, we discussed how the information provided by this combination of methods can be used to develop wildlife-friendly strategies that also maximize animal welfare, quality and environmental sustainability.

## INTRODUCTION

The use of natural grasslands for livestock production epitomizes conflicts of interests between landscape conservation and socio-economic activities (*Saberwal, 1996*; *Baldi, Albon & Elston, 2001*; *Treves & Karanth, 2003*), and gathering precise data for quantification of costs and benefits for both parties is challenging. A solution may lie in precision livestock farming (PLF), which has been proposed as a new and promising approach that allows animal welfare and economic productivity to be balanced with landscape conservation (e.g., *Berckmans, 2014*; *Kokin et al., 2007*). The main aim of this approach is to monitor continuously all the factors that might influence animal productivity and welfare to develop sustainable management strategies (see *Berckmans, 2006*; *Laca, 2009*; see examples in Fig. 1). Most of the tools employed for PLF include sensors, such as video cameras, accelerometers and pedometers (*Nadimi, Tangen Søgaard & Bak, 2008*; *Martiskainen et al., 2009*) that document animal behaviour (e.g., *Krohn & Munksgaard, 1993*; *Rushen, Haley & Passille, 2001*). PLF utilizes a combination of tools and methods to measure different variables from individual animals continuously and with high precision, and to process that information to help in the design of management strategies for livestock production systems (see examples in Fig. 1A). For instance, under intensive management, accelerometers attached on lambs can be employed to monitor feeding rate, maternal care, detect diseases such as mastitis in ewes, and also to predict the most profitable moment for weaning (see more examples in *Berckmans, 2006*; *Rutter, 2012*). However, the application of the existent tools for PLF can be challenging under extensive livestock management because this occurs on natural pastures which are large, heterogeneous and highly dynamic environments (*Wishart, Morgan-Davies & Waterhouse, 2015*; *Morgan-Davies et al., 2017*). Nevertheless, if the information is not only gathered through time, but also through space (i.e., spatially explicit), PLF tools provide data that could be also used to assess and minimize the impact of livestock on the landscape (*Misselbrook et al., 2016*), thereby maintaining livestock productivity (e.g., *Umstatter, Waterhouse & Holland, 2008*; *Umstatter, 2011*). Moreover, this data could be applied to maximize the reproduction and survival of livestock and wildlife by providing detailed information in time and space of the interactions between domestic and wild species. This would allow the design of an integrated management of wild and domestic fauna in rangeland ecosystems, where the conflict between livestock production and conservation can be high.

Livestock production in rangelands is characterized by low levels of human intervention, extensive paddocks with spatial and temporal heterogeneity, harsh climatic conditions and complex interactions between livestock and wild species. There is particular concern about interactions with wild species on extensive systems, mainly in relation to the loss of livestock to predators, and to forage depletion by wild herbivores (e.g., *Treves et al., 2004*). Examples of PLF technologies applied in rangeland systems typically focus on

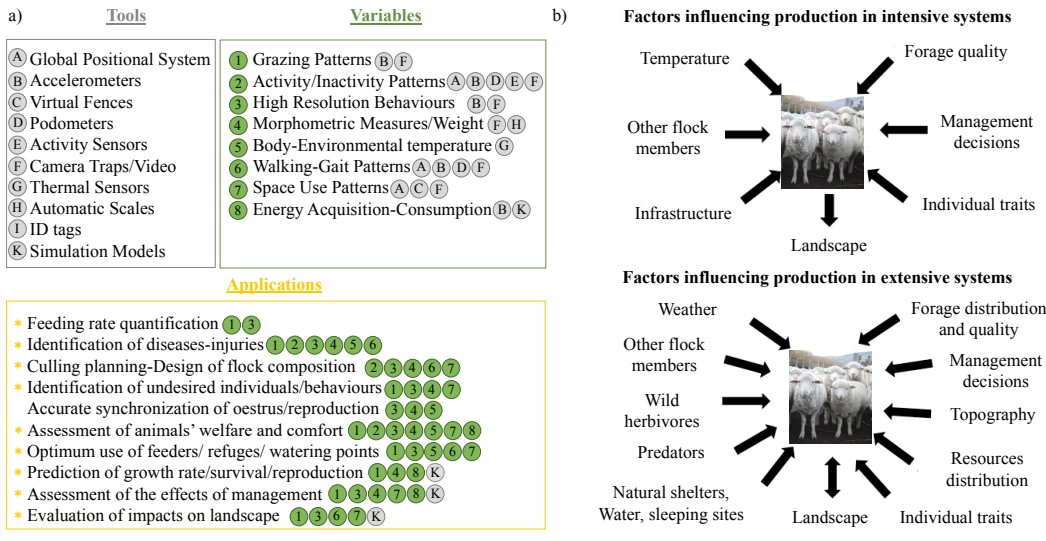

**Figure 1** **Precision Livestock Farming applications and potential factors influencing livestock production.** Examples of tools that are commonly applied in intensive systems of livestock production, variables that can be obtained with those tools and their application to accomplish different production aims (A); and factors influencing intensive and extensive livestock production systems (B). In (A), the tools are listed with capital letters (from A to K) and enclosed in a grey circle inside a grey box. Variables are listed with numbers (from 1 to 8) and enclosed in a green circle. The tool that could be used to obtain each variable is specified with the corresponding capital letter. All the applications of each variable and tools are enclosed in the orange box. Those applications highlighted with an asterix could be accomplished with our SMS approach. Also, the variables and tools that could be used for each application are denoted with the corresponding number and capital letter. Photo by: María Andrea Relva.

monitoring vegetation growth and herbage availability (e.g., *Schellberg et al., 2008*), with the majority dealing with virtual fencing (see review in *Umstatter, 2011*) and applied mostly to cattle. Importantly, although a few studies have sought to develop behaviour monitoring techniques for free-ranging domestic animals (e.g., *Ungar & Rutter, 2006*), they have been conducted in pens or small paddocks, without considering the different drivers of behaviour that occur under extensive production (Fig. 1B). The difference between both types of production systems is profound because, under extensive managements, livestock is exposed to the same variability in abiotic and biotic factors as wild animals (Fig. 1B). Among abiotic factors, the most influential drivers of animal movement and activity budgets are topography (e.g., *Kie, Ager & Bowyer, 2005*; *Dickson & Beier, 2007*) and temperature (*Patterson et al., 2009*; *Humphries et al., 2010*). However, these drivers have to be equated with the spatial and temporal distribution of forage quality and quantity, which are critical modulators of herbivore space-use (e.g., *Bailey et al., 1996*; *Johnson et al., 2002*; *Barraquand & Benhamou, 2008*). Beyond this, there is increasing interest in quantifying the effects of competition and predation risk on herbivore movement and behaviour (e.g., *Schuette, Creel & Christianson, 2013*), although these are not commonly assessed for livestock and are rarely included into management strategies. As for any wild animal these landscape drivers can influence individual reproduction, growth, survival, and movement

decisions, which ultimately impacts vegetation dynamics (*Morales et al., 2005*; *Morales et al., 2010*; *Nathan et al., 2008*).

The aim of this study is to develop a methodology to quantify continuously through time and space the following items: (i) detailed behavioural patterns, (ii) feeding rates, and (iii) costs related to landscape traits and missed opportunity costs associated to different behaviours. For this, we used Merino sheep in Patagonian rangelands as case study. We combined data from an animal-attached multi-sensor tag system (tri-axial acceleration, tri-axial magnetometry, temperature sensor and GPS) with landscape data from a Geographical Information System (GIS) to show how this combination of techniques can be used to assess feeding rate and costs from locomotion, thermoregulation, predation risk and competition experienced by livestock through time and space. We show results about the performance and potential applications of this methodological approach, which we named Spatial Multi-Sensor approach (hereafter, SMS approach), and then discuss its potential to design more sustainable and profitable management strategies for livestock and wild animals alike.

## MATERIALS AND METHODS

### Study site and landscape description

Data was obtained at Fortín Chacabuco ranch (41°0′46.36″S, 71°8′35.78″W) in November 2014 (for more details of the study site see Supplemental Information S1). The paddock where animals were kept is composed of different vegetation units including: (1) low productivity zones (i.e., areas with low forage production and a high proportion of bare ground); (2) riparian forests associated with a semi-permanent water stream; (3) native forest in the upper zone of the paddock, with more closed vegetation structure (mainly woody species which obstruct visibility); (4) shrublands, adjacent to native forests with lower vegetation; (5) grasslands located at the lowest areas in the paddock, with more dispersed vegetation; and (6) seasonal wet-meadows with high quality forage, but with two differentiated areas: (a) central and (b) peripheral meadow (Fig. S1.2 in Supplemental Information S1). For this paddock, we compiled a Geographical Information System including the following layers: (1) vegetation[1] (constructed as a supervised classification map with field data from cover of all plant species), elevation and slope (constructed from a Digital Elevation Map), predation risk (constructed by combining GPS locations of scats and footprints of the main predators in the area [i.e., pumas and red foxes], prey carcasses and vegetation type) and intra-specific competition data (built by using GPS location of sheep in that paddock). More details of this spatial data processing description are in Fig. S1.3, S1.4 and S1.5 from Supplemental Information S1.

### Tag description

We equipped three Merino sheep with *Daily Diary* (http://www.swansea.ac.uk/biosci/research groups/slam/slamtechnology/) (DD, http://www.wildbyte-technologies.com/; UK) and GPS devices (CatLog-B, Perthold Engineering, http://www.perthold.de; USA) for a period of 15 days. The sheep belonged to a flock of approximately 200 sheep. The DD recorded data from a 3-axial accelerometer, a 3-axial magnetometer, external temperature and pressure

[1] National Parks permit #1399.

at 40 Hz, among other data (see *Wilson, Shepard & Liebsch, 2008* for more details). These devices allowed us to infer behavioural and movement patterns of individuals with high spatial and temporal resolution together with relevant environmental data, such as temperature or pressure.

We attached two DD devices to each sheep to compare the results obtained from different parts of the animals; one device was attached to the back of the sheep's head (hereafter, head DD), and on the other one to the neck, attached to the GPS collar (hereafter, collar DD) (see Fig. S1.6 in Supplemental Information S1). Both DD devices where programmed to record data at 40 Hz (i.e., 40 data per second); and GPS devices where programmed to register location data once per minute.

To link DD signals with sheep behaviour, we first obtained a training data set that consisted of behavioural observations of each sheep equipped with both devices (DD + GPS). This data was collected by focal sampling of all tagged individuals on three different days (which resulted in a total calibration period of 15 h). We recorded the activity of each animal for periods of 5 min, with 1-minute breaks between observations. All behavioural observations were performed by the same person and the days selected to perform those recordings were included within the 15 days of sampling period. The six main behaviours identified visually were: Grazing, Resting-Rumination (hereafter Resting), Fast walk, Vigilance, Search, and Agonistic interactions (i.e., negative interactions among flock members). Within each grazing period, when possible, we identified when the animals clipped (bit) the vegetation (see details of all behaviours in Table S2.1 in Supplemental Information S2). Although we observed agonistic interactions among focal sheep, this behaviour resulted in very complex movement signatures that were hard to detect in the DD signal. For this reason, it was excluded from the list of behaviours considered in this study. In addition, vigilance behaviour included not only the interruption of activities under perceived predation risk but also the interruption caused by possible interactions with other herbivores. Finally, because sheep often ruminate while resting, we were not able to differentiate rumination from resting in the DD signal and thus considered them as one behaviour. The training data set was used to develop and validate the behavioural classification models explained below.

## Data processing and analysis
### Path reconstruction: spatially-explicit behaviours, costs and feeding rate
In order to link high resolution behavioural patterns with landscape information, we reconstructed individual movement paths using Dead Reckoning techniques (DR; *Wilson et al., 1991*; *Wilson et al., 2007*; *Bidder et al., 2015*). DR method requires the animal's initial location (e.g., Latitude and Longitude) and movement parameters such as velocity or proxies for speed such as data from dynamic body acceleration (e.g., Vectorial or Overall Dynamic Body Aceleration (VeDBA or ODBA, respectively), *Qasem et al., 2012*) together with heading data to derive each location with respect to the previous location (*Bidder et al., 2012*; *Bidder et al., 2015*). This method is based on vectorial calculation and can accumulate errors during trajectory estimations (see Fig. S3.1 in Supplemental Information S3). The optimal way to minimize path error is to combine

high resolution acceleration and magnetometer data with a lower temporal resolution GPS data to correct for system biases in the DR path (*Wilson et al., 2007*; *Shiomi et al., 2010*; *Liu et al., 2015*). The particular advantage of a GPS-enabled Dead-Reckoned path is that it can reduce GPS fix frequency (and thereby minimize battery power consumption) while deriving highly resolved movement paths that suffer minimally from GPS space resolution errors (see *Bidder et al., 2015*). Additionally, accelerometer and magnetometer data can provide additional information relating to behaviour and energy expenditure (see further sections below for more details) that cannot be obtained with GPS data alone.

To obtain unbiased and accurate DR movement paths, we first reconstructed movement paths using the TrackReconstruction package (*Battaile, 2015*) from R software (*R Core Team, 2016*). We then corrected these Dead Reckoning paths using two different approaches: (1) adding a linear bias correction (i.e., deterministic and conventional approach, *Wilson et al., 2007*; *Bidder et al., 2015*); and (2) using a Bayesian melding approach (*Liu et al., 2015*). We compared both approaches (see Supplemental Information S3) and selected the Bayesian melding approach, because, besides being the most accurate method, it also provided a measure of uncertainty around the estimation of the path (see Fig. S3.1 in Supplemental Information S3). After obtaining our unbiased DR paths, we thinned them from 40 Hz to 1 Hz, which represents a good compromise between resolution and computational demands. For each location on the DR corrected paths, we estimated the mean Vectorial Dynamic Body Acceleration (VeDBA, see details below) and mean temperature (extracted from the onboard sensor), and the information about elevation, slope, risk and competition levels, and vegetation from the maps described in Supplemental Information S1 (see details of this processing and a code example in Supplemental Information S3).

### Behavioural classification

To obtain high resolution behavioural patterns of sheep continuously through time, we constructed decision trees for each behaviour in the training data set using the "Behaviour Building" tool from Daily Diary Multi-Trace software (Wildbityes technologies, http://www.wildbyte-technologies.com/, UK). To accomplish this, we first separated the static (postural) and dynamic (movement) component of the acceleration data (e.g., *Wilson et al., 2006*), to estimate two variables: Head Pitch angle from the static component and Vectorial Dynamic Body Acceleration (VeDBA) from the dynamic component (*Gleiss, Wilson & Shepard, 2011*; *Qasem et al., 2012*; Fig. 2). Head pitch angle indicated if the animal's head was raised, lowered or in a neutral position; and VeDBA was taken as a proxy of energy expenditure associated with movement (e.g., *Qasem et al., 2012*; *Halsey, Shepard & Wilson, 2011*), which is obtained from the combination of the dynamic component of acceleration in the three axes (Fig. 2). Also, we estimated Activity Counts (AC) from raw acceleration data using the Daily Diary Multi-Trace algorithm, which is a measure of body acceleration within a time interval. We selected Head Pitch angle, VeDBA, AC, raw acceleration and raw geomagnetism data to construct our decision trees for all observed behaviours (see an example in Fig. 2).

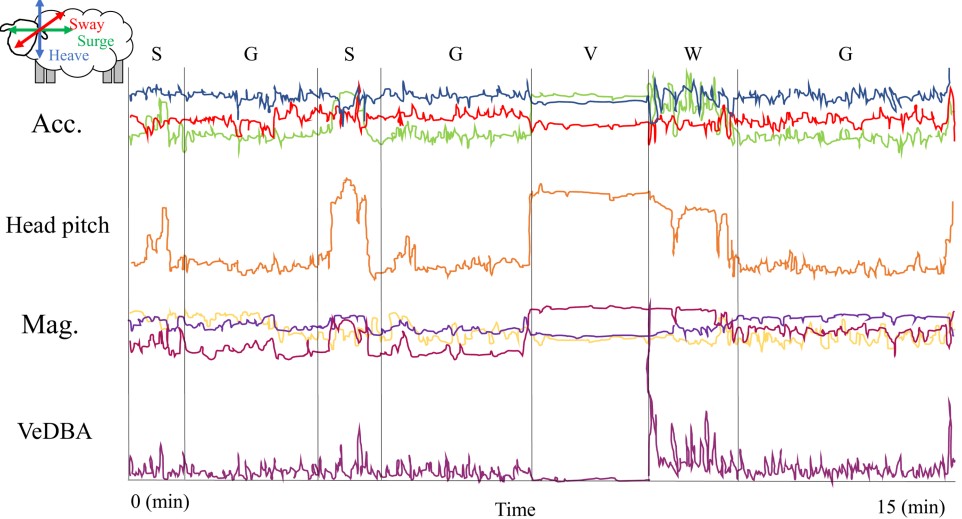

**Figure 2  A 15 min segment of the training data set with four behaviors (G, Grazing; V, vigilance; S, search; W, walk) performed by a sheep through time.** The behaviours observed in this 15-minute segment of the training dataset were: Search (S), Grazing (G), Vigilance (V) and Fast Walk (W). The top three traces show the three acceleration axes of the sheep's body derived from the Daily Diary attached to the back of the head (the red, green and blue lines correspond to the sway, surge and heave acceleration, respectively (see the arrows on the figure of the sheep in the top left-hand corner)). The orange line shows the head pitch while the pink, violet and yellow-ochre lines correspond to the $x$, $y$ and $z$ axes from the magnetometer. The bottom line (magenta) is the Vectorial Dynamic Body Acceleration. For instance, we can see that when the animals are grazing, head pitch is always lower than the periods where is searching, running or vigilant. Also, when animals are searching, the magnetometer signal is very noisy, with frequent changes in the values from the three axes. In contrast, when animals are vigilant, the signal from the magnetometer is constant and also the VeDBA values are significantly lower and more or less constant. And when animals are running, the main trait is the significantly high VeDBA values and a less variable signal from the magnetometer axes.

After selecting the above-mentioned variables, we randomly divided the training data set into two subsets; one to construct and the other to validate the classification model (see *Nathan et al., 2012*). From the fraction used to construct the model, we estimated the mean, maximum and minimum values of all variables for each behaviour in the training data set, in order to obtain the threshold values used to build the decision tree. Then, we fitted the trees to the training data set to assess the accuracy in the classification (see Fig. S2.1 in Supplemental Information S2). This accuracy was evaluated by quantifying the proportion of observations that were correctly classified, and also constructing a confusion matrix to assess which behaviours were misclassified. After obtaining a high accuracy in the prediction of decision trees for each behaviour in the training data set (see Supplemental Information S2), we applied them to the entire data set for all individuals. To match the temporal resolution of DR paths, we thinned behavioural data from 40 Hz to 1 Hz. As result, we had behaviours associated with every location along the movement path, and in the case of bite behaviour we also estimated the number of observations per second that were classified as bite, to obtain a bite rate for each location (i.e., number of bites in a second). Particularly for bites, we used a 7-hour period of grazing from our training data

set to extract the exact times where we detected bites and compared these with the times the algorithm found a bite. Finally, we compared the classification accuracy obtained with the data from the Daily Diary attached to the collar.

### Accurate feeding rate (grazing intensity)

After we had identified short duration behaviours such as bites with high accuracy, we quantified the number of bites per second to obtain feeding rate through time for each individual. Also, we associated a bite rate value for each location along the path, to have a spatially-explicit pattern of intake rate.

### An approximation of costs

Some behaviours, such as vigilance for predators or interference competition, interrupt grazing and thus decrease grazing efficiency (i.e., when the number of interruptions increase, the amount of food ingested in a period of time decreases). Within this context, the behavioural cost associated with perceived predation risk and competition can be assessed by estimating individuals' vigilance levels (i.e., time spent on vigilance). After obtaining vigilance levels through time and space, we related this variable with information from risk and competition maps, to assess the regions in which this indirect cost is higher. Also, based on the observation that movement is the main modulator of energy expenditure in vertebrates (e.g., *Karasov, 1992*), and that body acceleration is highly correlated with energy expenditure (e.g., *Wilson et al., 2006*; *Halsey et al., 2009*), we estimated VeDBA values for each behaviour performed by sheep along their movement paths. Then, we correlated VeDBA values with landscape traits such as elevation and slope to approximate movement costs (expressed in terms of dynamic acceleration) imposed by topography. We additionally extracted the temperature values from DD sensors to obtain the approximate environmental temperature that animals might face through time and space that could be used to calculate thermoregulation costs.

## RESULTS

### Behavioural classification

The most frequent behaviours in the dataset were grazing and resting, followed by searching behaviour (Fig. 3). Although we obtained a relatively high classification accuracy with both the head and collar DD, with most behaviours being classified with an accuracy in excess of 75% (Table S2.2 in Supplemental Information S2), the head DD showed consistently higher classification accuracy not just for the most common behaviours, but also for rest of the behaviours. This resulted in the following classification accuracies for each behaviour: 93% for Grazing, 87% for Searching, 97% for Fast Walking, 79% for Vigilance and 75% for Resting in head DD. Our algorithm classified all behaviours consistently, with a low proportion of misclassification (see Table B3 in Supplemental Information S2). Interestingly, in the case of bites, we obtained a perfect match (100% for the head DD and 97% for the collar DD) between the time when we observed a bite and the times when the algorithm found a bite. The estimation of the mean (±standard deviation) bite rate for the

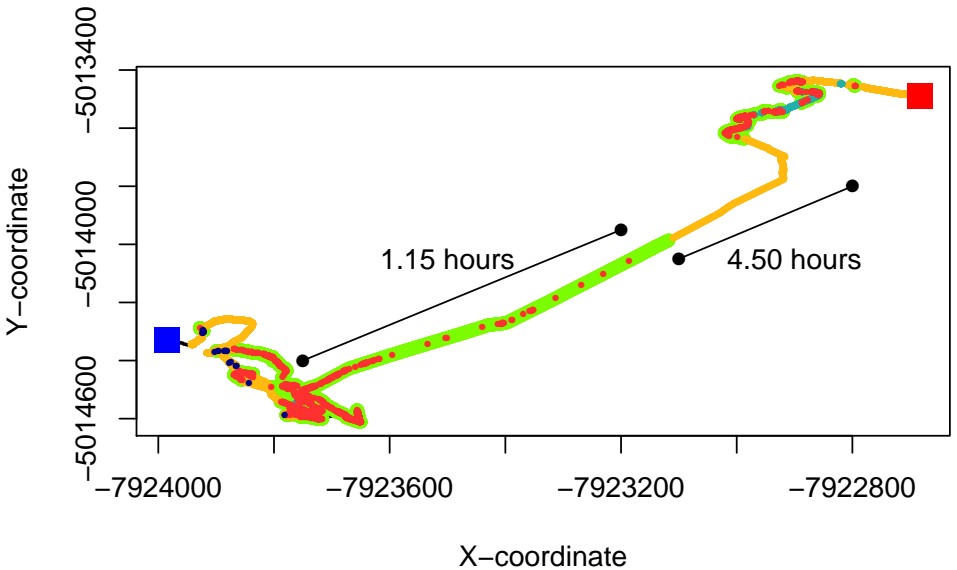

**Figure 3** **GPS-corrected dead-reckoned path showing space-associated behaviors from one sheep over 24 hours.** The red and blue squares represent the start and end points of the movement trajectory. Grazing points are colored in green, resting in orange, search in red, fast walk in light blue and vigilance in blue. The uncolored parts of the paths (black line) are location that could be classified in any of the behaviors considered in this work. The straight lines adjacent to the path show the time involved in a grazing bout and a diurnal resting period, where the individual seems to move at a very low speed.

observed bites and the classified bites resulted in 1.22 (±0.52) observed bites per second and 1.43 (±0.63) classified bites per second.

Through this behavioural classification, we were able to obtain the proportion of time that animals spent performing different behaviours. For instance, we estimated the average (±standard deviation) proportion of time that animals spent grazing (0.36 ± 0.04), resting (0.25 ± 0.07), searching for food (0.31 ± 0.04), being vigilant (0.02 ± 0.02) and running (i.e., fast walking; 0.06 ± 0.03) during the sampling period. Moreover, by combining this classification with dead reckoning, we assessed how the allocation to different behaviours change through time and space (see Fig. 3 and Figs. S4.1 and S4.2 in Supplemental Information S4). This also allowed us to observe which areas within the paddock elicited higher searching times for the animal. This variable could be used to infer patch quality, because areas with lower search times could indicate higher abundance of resources or a less heterogeneous distribution of resources (*Gross et al., 1995*).

### Accurate feeding rate

The high accuracy in the classification of grazing behaviour and bites allowed us to estimate forage consumption rate for each animal through time and space, which roughly translate into grazing intensity across a movement trajectory (Fig. 4A). This also allowed us to detect which vegetation types and landscape regions are exposed to higher grazing levels, and to assess daily feeding dynamics of bites (Fig. 4B). Our SMS approach showed how grazing intensity varied radically with space (Fig. 4A) within and between individuals (Fig. S4.3

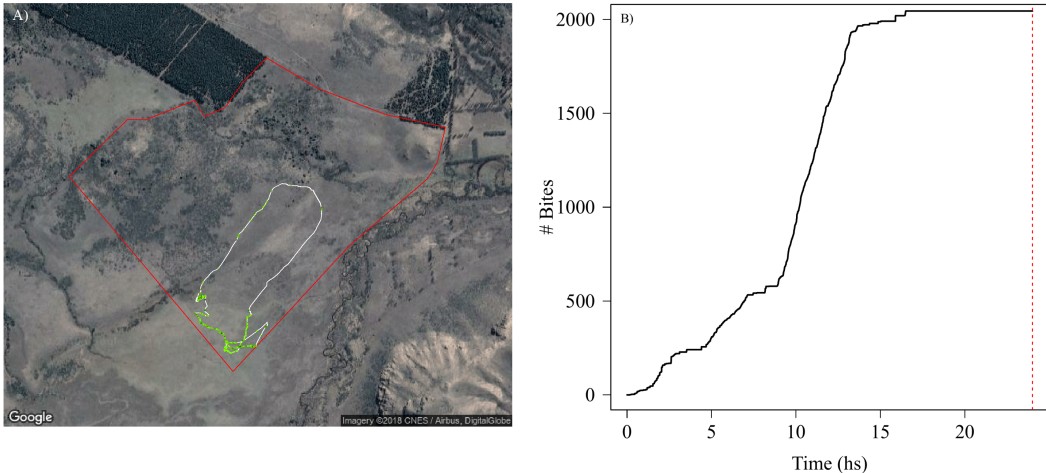

**Figure 4  Spatial and temporal patterns of bite dynamics of a sheep during 24 hs.** In (A) is the reconstructed movement path of a sheep within the paddock (delineated by red lines), color-coded according to estimated bite rate: The white solid line in the movement path shows parts of the trajectory where grazing did not occur while different tones of green represent different bite rates that ranged between 1 and 4 bites per second (i.e., darker tones indicate higher bite rates). We can observe that higher bite rate values occurred on meadows where the vegetation has less fiber content and higher quality. In (B) is shown the cumulative bites over time through the day for the same period of (A). The red dashed line indicates midnight. We can detect periods where animals do not feed and periods of high intake rate along the day. Map data: Google, DigitalGlobe.

in Supplemental Information S4). For instance, we could detect periods where animals do not feed and periods of high feeding rates (Fig. 4B), and the number and duration of non-grazing periods (represented by the flat parts of the curve in Fig. 4B). Although the movement paths might be similar between animals, their feeding behavior can be quite variable, with animals grazing more intensively through the landscape than others producing marked differences in the amount of food ingested in the same period of time (Figs. S4.3.1, S4.3.2 and S4.3.3 in Supplemental Information S4).

## An approximation of costs

By combining data from risk and intra-specific competition with behaviour classification and dead reckoning paths, we show that is possible to assess vigilance costs across areas with different levels of perceived predation risk or competition. For instance, we could see that areas with high predation risk showed higher vigilance levels (Fig. 5A) and areas with high intra-specific competition showed the lowest vigilance levels (Fig. 5B). This pattern might indicate that during the sampling period, perceived predation risk could be the main factor influencing vigilance levels. Moreover, we could assess how vigilance varied between individuals and days, to test if a particular factor (i.e., risk, inter or intra specific competition) represents higher costs for a particular individual or during a certain period of the year or within a particular paddock.

Furthermore, using this data we assessed how vigilance varied through space (i.e., along movement paths, Fig. 6A) and through time (Fig. 6B). For instance, there is a large

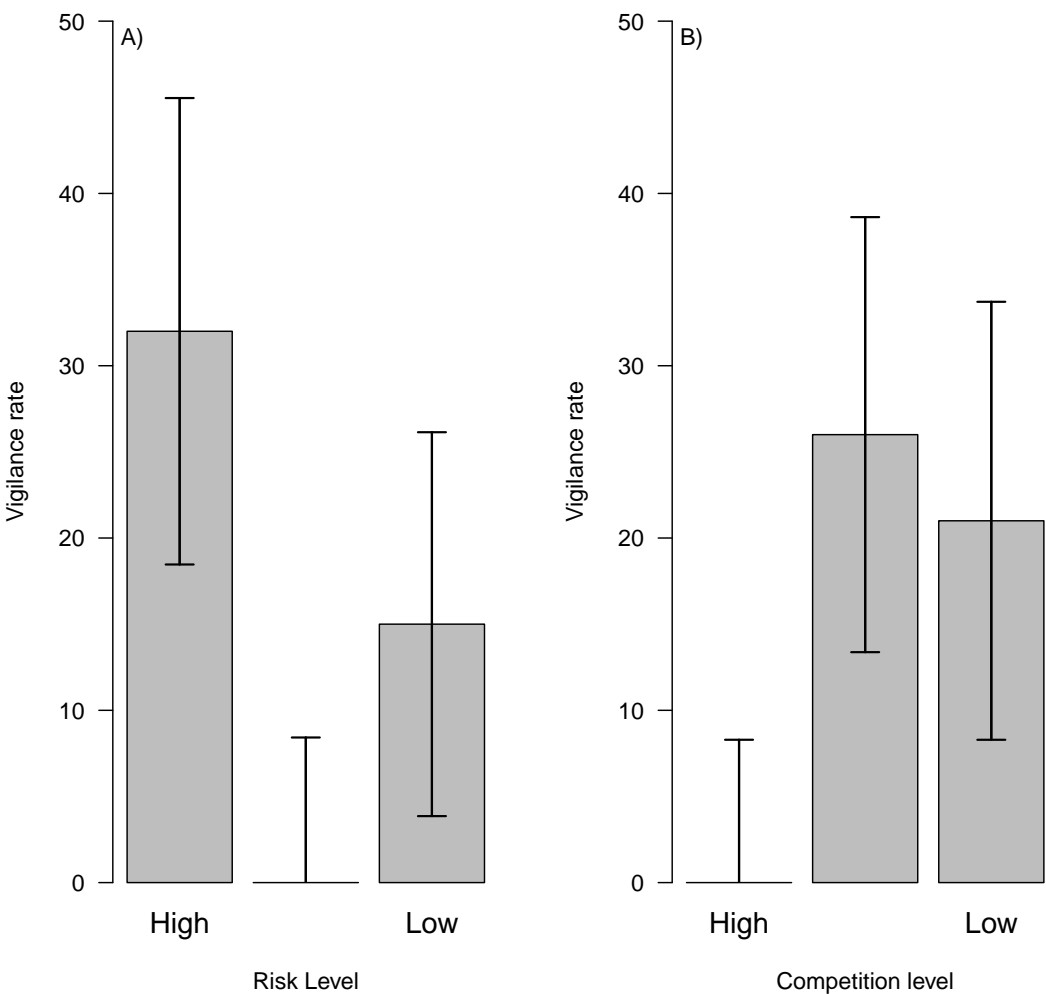

**Figure 5  Vigilance rate related to risk level and intra-specific competition.** Vigilance rate (i.e., number of interruptions per second) related to risk level (A) and intra-specific competition for food (B) across the landscape. The bar height represents the mean value and the error lines represent the standard deviation

proportion of the movement paths where animals did not show vigilance behaviour (e.g., white solid line in Fig. 6A), but there were certain periods of the day that showed high vigilance levels (for instance, between 14:00 and 17:00 in Fig. 6B, when the individual bypassed an area with very dense and closed vegetation, where perceived predation risk is higher). In the same way as with bite rate, we were able to detect variability in the spatial and temporal patterns of vigilance behaviour among individuals and between days (Fig. S4.4 in Supplemental Information S4). For instance, although individuals could show similar movement paths and some coincidence in the areas where vigilance occurred, vigilance levels could vary among them (Fig. S4.4 in Supplemental Information S4). The differences observed among the individuals tracked here provide evidence that this approach could be used to detect individual variations in perceived predation risk when applied to several individuals within a flock.

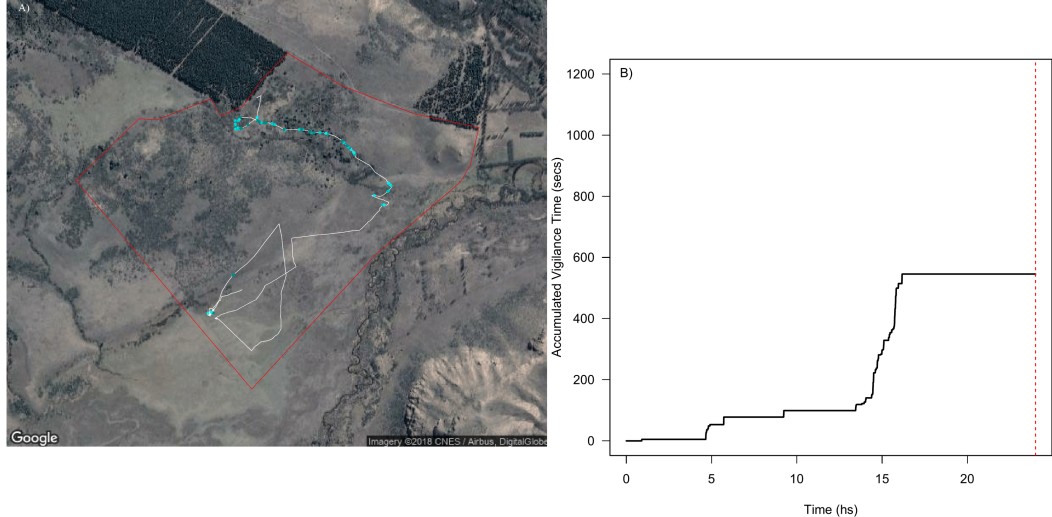

**Figure 6  Spatial and temporal patterns of vigilance dynamics of a sheep during 24 hs.** Movement path of a sheep over 24 h with vigilance rate linked to location via color-coding ((A) the white solid line represents parts of the trajectories without vigilance and the blue dots indicate different vigilance levels (darker values indicate higher vigilance rate)) and cumulative number of seconds spent being vigilant through the day (B). The red dashed line from (B) indicates midnight. Map data: Google, DigitalGlobe.

We also used VeDBA and temperature for each individual to assess the spatial dynamics of movement (a proxy for movement-related energy expenditure—*Halsey et al., 2009*—Figs. 7A and 7C) and thermoregulation costs (Fig. 8). We showed that variable VeDBA levels within paths can be related to landscape traits that could affect movement costs, such as altitude or slope (Figs. 7B, 7D). Also, our approach allowed us to detect different responses of individuals to topography. For instance, in Fig. 7, we can see that two individuals with similar movement paths (i.e., operating on similar slopes) exhibited different movement costs particularly when slope angles are higher.

## DISCUSSION

The combination of high resolution data from multi-sensor tags and landscape data proposed here (the SMS approach), would appear to be a useful tool with which to assess livestock behaviour and to quantify the effects of the different factors affecting extensive livestock production. For instance, through the SMS we were able to quantify feeding rate continuously through time and space with high accuracy, which is extremely valuable data that can be used to estimate animal production and also grazing intensity within landscapes (e.g., *Galli et al., 2017*; *Galli et al., 2011*). Moreover, we could assess the effects of resource heterogeneity (inferred through search times), and the potential costs associated with predation risk, competition, thermoregulation and movement over complex topography. Although agonistic behaviours are not easily observed and sometimes difficult to define, we encourage future users of the SMS-approach to focus on gathering more observational data on this behaviour. The possibility of detecting agonistic interactions among flock

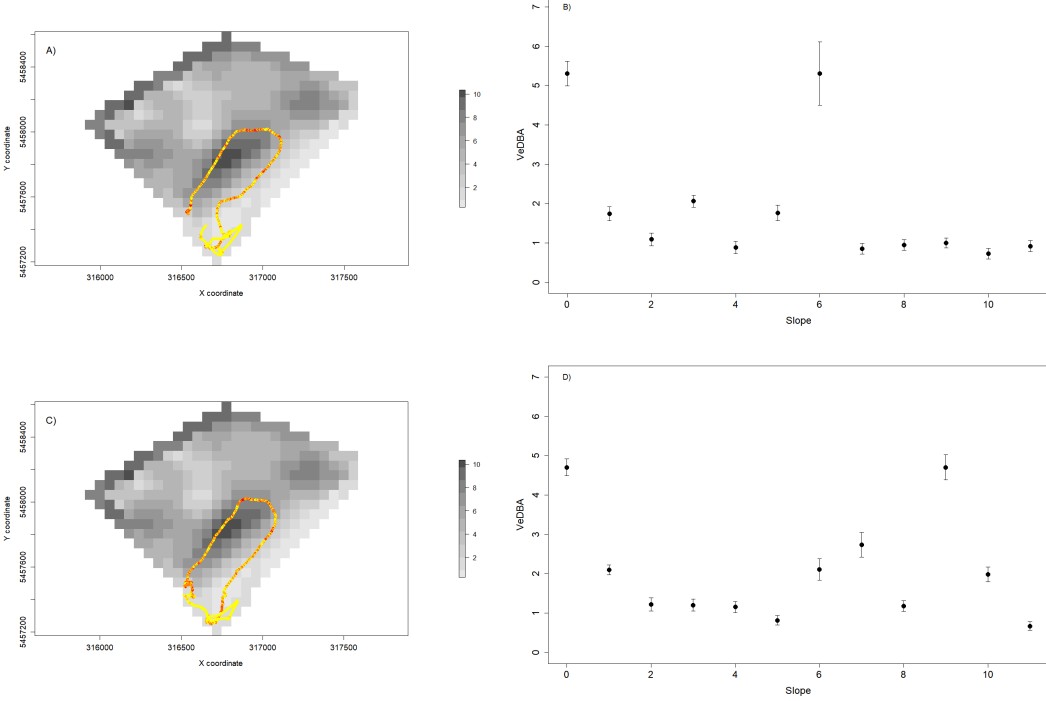

**Figure 7** **Relationship between VeDBA values and paddock slope for two sheep during a 24 hs period.** (A and C) show the daily path of two sheep colored by VeDBA values (i.e., yellow represents low values and red represent high values); and the underlying landscape is represented by a raster constructed with slope data. (B and D) show the relationship between the mean values (±standard error) of VeDBA and landscape slopes for each sheep. As we can see, although the space use is similar, the energetic costs associated to movement show some differences, particularly for high values of slopes.

members or between livestock and wild herbivores would contribute to quantify more accurately the effects of competition.

Our SMS approach has four main advances compared to previous methods: (1) we were able to identify and classify not only grazing and resting (or active versus non-active periods; e.g., *Müller & Schrader, 2003*; *Umstatter, Waterhouse & Holland, 2008*), but also more detailed information such as bite rate, vigilance and food search under extensive conditions; (2) the quantification of different activities through time and space allowed us to link behaviour to different environmental traits such as perceived risk, competition and topography; (3) the quantification of potential energy gains (through bite rate) and costs (associated to missing feeding opportunities, movement, and temperature) that can be used to estimate energy balances and predict individual growth, survival and reproduction (e.g., *Belovsky, 1986*; *Moen, Pastor & Cohen, 1997*) through allometric equations available in the literature; (4) the ability to detect individual variability in key processes, such as forage consumption rate and vigilance costs, that affect production and sustainability. For instance, if we only have information about the time that individuals spend grazing—as is the case in most previous studies—we cannot assess if animals have the same intake rate or are grazing on the same vegetation patch. This is partly because two different

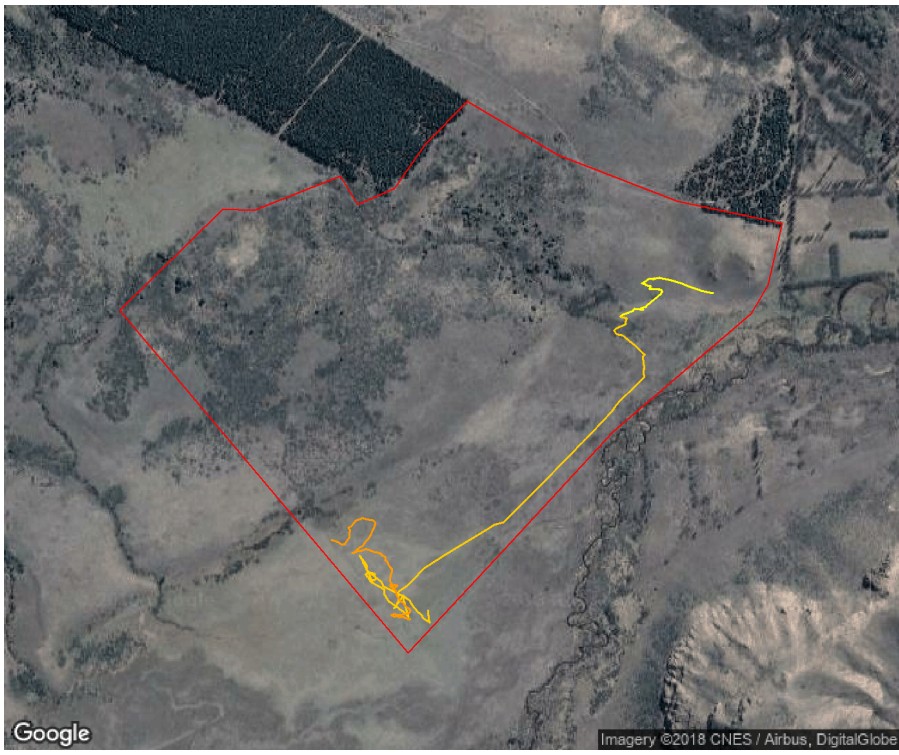

**Figure 8  Movement path showing the approximate environmental temperatures experienced by sheep during 24 hs.** The color-coded movement trajectory of a sheep over 24 h (high temperatures are shown in darker colors) so that environmental temperature can be linked to time and space. For this day, temperatures ranged from 10 °C registered during early hours to 23 °C registered around noon. The red solid line shows the paddock fence. Map data: Google, DigitalGlobe.

individuals may spend a similar amount of time grazing but have significantly different intake rates, with concomitant impact on vegetation. We note also that our methodology allows us to infer if the variability among individuals is because they moved differently, grazed differently or because they reacted differently to a particular landscape feature, such as predation risk or competition. Considering that the aim of this study was to present a methodological approach that could be applied as a PLF tool in rangelands, future efforts should be focused on the development of hardware and tools that allow movement and acceleration data to be collected remotely and user-friendly platforms to present the results. For a more theoretical and ecological point of view, we would also encourage future research to apply this technique in a larger number of animals to provide inference and results about how livestock react to different factors under different management.

## Potential management and ecological applications

As with intensive systems, assessing continuously detailed behavioural patterns of livestock could be used to develop economically and ecologically sustainable management strategies which are compatible with landscape conservation (*Delcurto et al., 2005*), and also to evaluate the effect of a certain management strategies or to orient management decisions

(see examples in *Laca, 2009*). For instance, the SMS would allow users to assess individual differences and to associate them with intrinsic characteristics of individuals, such as age, sex and social status (*Hamel et al., 2009*). This association could help to manage livestock by grouping individuals in order to maximize production and minimize their impact on the landscape (e.g., *Di Virgilio & Morales, 2016*). Also, our approach could reveal which individuals are more exposed to direct and indirect effects of predation risk (*Lima & Bednekoff, 1999*), which individuals feed sub-optimally, and which show unwanted behaviours or any particular movement pattern that could indicate injuries or diseases. This information can be used to improve livestock welfare and minimize the impact of animals on the landscape. This relevant information could be used to improve management decisions and to allow a dynamic adaptation of management decisions according to flock composition and landscape heterogeneity. Some of these decisions might include determining flock composition according to behavioural patterns to maximize production (see references in *Bailey, 2004*). This could be achieved, for instance, by minimizing the negative effects of inter-specific interactions that affect grazing efficiency, such as using guardian dogs or changing flock composition by selecting higher proportion of less sensitive animals to predation risk. But also, by knowing which areas are most frequently used for different behaviours, livestock producers could select certain regions in the landscape to allocate different resources such as watering troughs, food supplements or even predator deterrent lights in known sleeping areas.

One of the most interesting applications of our SMS approach is that it has the potential to allow an integrated management of livestock, vegetation and wildlife. By constructing predation risk maps and inter-specific competition maps (if we consider the space use of wild herbivores) and assessing how animals react to these landscape traits, livestock producers could consider how wildlife can be included into their management strategies as another variable that can be managed. For example, managers could estimate resources overlap and overall grazing intensity along the paddocks to adjust the stocking rates to sustainable levels. The SMS would also allow the estimation of the effect of inter-specific competition on livestock grazing efficiency in different paddocks, and the identification of less sensitive individuals to inter-specific competition or to wild herbivores species that affect livestock grazing.

Similar objectives could be achieved in relation to predators and the well-known human-carnivore conflict (*Treves & Karanth, 2003*). There is a particular concern about the lethal effects of predation, with the effects being readily quantifiable (see *Miller, 2015* and references therein). However, it is known from wild animal studies that what is termed 'the landscape of fear' (*Laundré, Hernandez & Ripple, 2010*; *Gallagher et al., 2017*) modifies the behaviour of prey without lethal effects. Those non-lethal effects have profound implications for energy dynamics, modulated via behavioural responses such as space use and vigilance levels. Many of these indirect effects are similar to those related to competition because vigilance and agonistic interactions result in a reduction of grazing efficiency (e.g., *Lima, 1998*; *Manning, Gordon & Ripple, 2009*) and change livestock space use patterns (e.g., *Brown, 1999*). Our results show that we could develop a landscape of fear by considering vigilance behaviour combined with predation risk map to infer which regions are more

costly due to predation risk. Also, we could evaluate if the landscape of fear is similar for most individuals, and if it changes through time and across paddocks to optimize the use of different strategies to mitigate carnivores-human conflicts.

Lastly, this approach could be extended and used on different domestic and wild herbivores, by using allometric and standard equations to convert behavioural patterns into energy dynamics (*Brosh et al., 2014*) and the data obtained could be used for the development of simulation tools to predict production and sustainability (e.g., *Dieguez Cameroni & Fort, 2017*). For instance, after obtaining detailed information of bite rate it should be possible by using the nutritional values of plants and known bite size to quantify the amount of energy gained by animals (e.g., *Osuji, Gordin & Webster, 1975*; *Wilmshurst, Fryxell & Bergman, 2000*). Following this step, the behavioural costs of risk and competition could be approximated by converting the time when animals interrupted behaviour into bites not performed, to express these missed opportunities as energetic costs. Moreover, for movement costs it is possible to use VeDBA values (or similar proxies) and the available regressions of its relationship with energy expenditure (e.g., *Weippert et al., 1999*) to express movement costs in energetic units (e.g., *Wilson et al., 2013*). For the thermoregulation costs, we could use the data from temperature loggers and the equations of thermoregulation according to environmental temperature (*Jensen, Pekins & Holter, 1999*; *Garrot et al., 2003*). The possibility of quantifying the energetic dynamics of herbivores could be used to understand how animals balance their trade-offs and allocate time to different behaviours according to their energetic status and demands which ultimately determine population dynamics (*Shepard et al., 2008*; *Morales et al., 2010*; *Brown et al., 2013*).

## CONCLUSION

For precision livestock farming, each individual is considered a complex system, which is unique and changing adaptively through time and space (*Berckmans, 2006*). There is a strong consensus about the conditions that need to be met for a method to be considered as PLF tool: (1) it should allow different animal traits to be measured continuously and with high resolution, such as weight, activity, behaviour, and feeding rate, among others; (2) all measures and outcomes should have high precision and accuracy to allow robust predictions about how animals will respond to different scenarios where the main processes affecting livestock production change; and (3) the measures can be processed through general algorithms that should be available and extended easily to other systems. We have shown that the SMS approach, combines different methods to acquire and process data that have been successfully implemented in different disciplines and for different purposes, and it therefore meets all these conditions and can be applied on extensive systems with high accuracy.

We wish to highlight that our SMS approach, besides allowing managers to maximize animal production and long-term sustainability of resources, could provide short-term income by contributing to wild species conservation. At present, livestock management strategies that tend to minimize the impact on wildlife and focus on resources conservation

have the potential to provide economic rewards for farmers because they are considered as environmentally- and wildlife-friendly (e.g., through eco-labels; see review in *Treves & Jones, 2010*). Lastly, although our approach could be applied in intensive systems, it is particularly relevant for rangelands, where the application of PLF techniques is very challenging (see Fig. 1). Considering that rangelands represent the 30–40% of the ice-free terrestrial surface and the 91% of the surface devoted to livestock production (*Reid, Galvin & Kruska, 2008*), our SMS approach has the potential to be applied at a global scale.

## ACKNOWLEDGEMENTS

We thank The Nature Conservancy (TNC) and S Gary for allowing us to conduct field work at Fortín Chacabuco ranch; to F Montenegro and N Rodríguez for all the assistance and invaluable help; to G Iglesias for their support and to CONICET. Also, we thank to our colleagues from "Grupo de Investigaciones en Biología de la Conservación (GrInBiC)" for all their comments that improved the previous versions of this manuscript; and to our colleagues from "Swansea Laboratory of Animal Movement (SLAM)" for all the technical and methodological assistance.

### Funding
The authors received no funding for this work.

### Competing Interests
The authors declare there are no competing interests.

### Author Contributions
- Agustina di Virgilio conceived and designed the experiments, performed the experiments, analyzed the data, contributed reagents/materials/analysis tools, prepared figures and/or tables, authored or reviewed drafts of the paper, approved the final draft.
- Juan M. Morales, Sergio A. Lambertucci and Emily L.C. Shepard contributed reagents/materials/analysis tools, authored or reviewed drafts of the paper, approved the final draft.
- Rory P. Wilson conceived and designed the experiments, contributed reagents/materials/analysis tools, approved the final draft.

### Animal Ethics
The following information was supplied relating to ethical approvals (i.e., approving body and any reference numbers):

Our research was conducted on non-regulated animals. The legislation of the site where we conducted our work did not require any ethical approval. Moreover, there is not a specific institution that regulates the manipulation of domestic animals. Note also, that the manipulation of the sheep included fitting GPS collars and multisensor tags to them externally.

## Field Study Permissions

The following information was supplied relating to field study approvals (i.e., approving body and any reference numbers):

The vegetation data used in this work to construct vegetation maps is shared with another project and researchers, for which we have applied for permission to National Parks Administration (Permit number 1399).

## Data Availability

The raw data and R code are provided in the Supplemental Files.

## Supplemental Information

Supplemental information for this article can be found online at http://dx.doi.org/10.7717/peerj.4867#supplemental-information.

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
