# Peer review of "Multi-dimensional Precision Livestock Farming: a potential toolbox for sustainable rangeland management"

_PeerJ, doi:10.7717/peerj.4867_

## Round 0.1 · original submission · Major Revisions

The two referees, who are expert in the field, are quite positive overall about the paper. I concur. The manuscript is very well presented and structured and of notable academic and applied value, taking a systematic approach that undoubtedly provides valuable information for managers seeking to manage landscapes more effectively, and an analysis and discussion that is also relevant for grazing management.

However, the referees see many areas that could be improved, and consequently have provided detailed comments. I have looked over their feedback with respect to the original submission, and see merit in all they suggest. I therefore recommended a major revision, with attention given in the response letter to how you have responded to each specific point.

Reviewer 1 ·

Basic reporting

The article is clearly written, and it is an important topic to study. The introduction and methods are clearly presented, although more details in the introduction would be beneficial (see later comments). The results section is a bit more difficult to follow, and there are some mistakes in the labelling of some of the figures. The discussion is however more of a series of the potential applications of the findings, whilst discussion on the method used would have been also interesting.
The raw data have been supplied, and, in places, maybe could have been included in the manuscript as figures.

Experimental design

The experimental design is clearly explained, although in places, more details would be useful. As it stand, it is difficult to really understand how many animals in total were involved, how long they got the sensors on, how many observers during the training set period, how long after the initial training session were the data recorded for. Also, there is no clear presentation on how the comparison and correlations were made – which stats package, which tests? It would be helpful to know this information .
You also need to explain why 3 sheep are enough for the experiment.

Validity of the findings

The results do not appear to present everything – I would have liked to see some information on the training dataset results. Also, a lot is inferred on only 3 sheep – so is it a case of lots of data for not many sheep, which is a trade off? This should be shown better.
It is not clear if the approximation of costs is purely based on energy costs, this should be explained.
In the discussion, I think that a paragraph detailing why 3 sheep are enough to infer the results is necessary. Did other studies have so few animals? Also, what about seasonal variation, animal variation, would it have been useful to rotate the 3 collars among different individual to obtain more data? This should be discussed. I also feel that lines 306-319 should be in the introduction, not the discussion. I also think that the section on management & ecological applications should be called ‘potential management & ecological applications’. In that section, the affirmations should be more cautious, using ‘could’ instead of ‘can’.
I am not sure if it is appropriate to include references in the conclusions.

Additional comments

More detailed comments below:
Title: I suggest you say: Multi-dimensional PLF – potential toolbox for sustainable rangeland management.
Introduction. I would add the section in the discussion (306-319) in the introduction
Line 48 – more definition of PLF would be good
Line 61 – more reference to this fact would be good
Line 71 – reference on PLF with wild species (e.g. GPS on deer, etc.). I would also add some reference to wild species monitoring.
Perhaps add a section on conflicts between the livestock and wild species would be useful.
Line 94 – Merino sheep – why sheep? Perhaps explain that there are few studies with sheep – most of PLF at present is on cattle or dairy or monogastrics.
M&M:
Line 108 – what do you mean by 1) low productivity zones?
Line 115 – how was the predation risk defined? Please explain a bit.
Line 118 – how many sheep in the paddock in total? Only 3 sheep on 90 ha? If you had more sheep, how did you choose them? What age were they? Were they naïve to the land? Were they related to each other, or did they come from the same original flock? All of these are important because of the gregarious nature of sheep.
Line 129 – how long did you measure the data after the initial training? This is not explained.
Line 131 – how many observers during the training data period?
Line 144-145 – I would reference Umstatter et al. 2008, who used similar approach.
Line 178 – competition level – how did you assess these?
Line 217 – how do you obtain intake rate from bite rate? Formulas exist in the literature (e.g. see Illius & Gordon in the late 90s who worked on these issues)
Results:
Line 239 – high classification accuracy – what is from the training set? You should precise this.
Line 261 – searching time vs less heterogenous distribution of resources – perhaps you should reference this.
Lines 268-269-271 – Figures 4, 4a, 4b – these are not in the figures list – it looks like there has been a switch between the figures, they do not correspond to the text.
Line 273 – ‘some animals’ how many? You only had 3 sheep.
Line 277 – predators – what sort of predators do you have in Patagonia? It would be nice to give some examples for the international audience.
Line 295 – ‘some individuals’ – again, which ones out of the 3?
Discussion
Line 330 – I would also cite Umstatter et al (2008) who looked at different behaviours and activities with sheep (resting, ruminating, moving, grazing).
Line 368 – I would say ‘could be that it allows…’
Line 392 – I would say ‘could’ instead of ‘can’

·

Basic reporting

Language is overall clear and professional and complex ideas are explained with clarity and detail.

Abstract discussion. The first line talking about how the method could determine “variability between individuals” is more methodological than discussion. It should either be related to the discussion more explicitly, moved to the methods / results or removed.

Figure 1a. Could be improved by explicitly linking the tools to the variables and applications. As it is, it doesn’t add much more than is already in the text.

Background, it would be nice to have a short description of how PLF is defined.

Welfare issues – for me it is not clear why the method would improve animal welfare explained why this relates to welfare? Presumably by helping reduce predation or detect lameness etc? It would be nice to have this clarified in the discussion.

Experimental design

The experimental design is robust and detailed. The authors have developed a methodology that extracts an impressive range of variables on animal behaviour and landscape features from the data.

Validity of the findings

The findings are valuable for identifying different animal behaviours that may help improve management and the SMS system would appear to offer great promise for monitoring and managing grazing herds in extensive systems.

Discussion and conclusions are well related to the methods.

Additional comments

The analysis is well presented, and highly relevant for managers seeking to improve productivity while managing grasslands and wildlife conflicts more effectively.

A few minor comments line by line are as follows:

Line 47: Awkward wording. Suggest rephrasing “quantification of detriments….”
Line 55: Suggest inserting word - document “animal” behaviour
Line 66: Suggest being more explicit about the ability of the method to reduce human wildlife conflict with both predators and wild herbivores
Line 96: What GIS system? Describe the source.
Line 139: Agnostic interactions. Please explain what this means
Line 277: “from on” both words are not needed
Line 307: “the” should be corrected to “there”
Line 348: Missing word – ecologically (?) management
Supplementary material S1, Line 24: Pow should be paw

---

## Round 0.2 · Minor Revisions

Two of the original reviewers have seen the revised paper, and both are happy with outcome of the refereeing process. They and I feel the paper adds clear value to the discussion on balancing agricultural productivity with reduced environmental damage from food production. Well done.

There are a few minor but useful revisions suggested by both referees, which I would ask the authors to undertake before a final version can be accepted.

Reviewer 1 ·

Basic reporting

This revised manuscript now reads very well, and is a very valuable piece of research. I would only suggest adding a precision in the title of Figure 2: "A 15 min [...] with four behaviour (G=Grazing, V=vigilance, S=search, W=walk) performed by a sheep through time".

Experimental design

The added details in the methods clarify it perfectly.

Validity of the findings

The revision in the discussion and results are well done. The changes in the conclusions are also fine.

Additional comments

I have few very minor comments:
-line 66: It is Morgan-Davies (not Morgan-davies)
-lines 108-112 - I am not sure these lines are necessary.
-line 131 - I would delete 'For' and rewrite the sentence as: More details [...] description are in Figs. [...].
-line 155 - I would add: The 6 main behaviours
Line 157 - change 'inside' to 'within'
Line 198 - I would put VeDBA in full letters at this stage.
Line 358 - I would change: ...sometimes difficult to be defined
Line 361 - I would change 'will' by 'would'
Line 384 - I would suggest "For a more theoretical and ecological point of view, we would also encourage future research to apply this technique..."
Line 479 - applied at a global scale.

·

Basic reporting

No comments

Experimental design

No comments

Validity of the findings

No comments

Additional comments

The updated manuscript is much improved, and I have no further substantive comments.
The manuscript was in good shape originally, but the new edits provide valuable context to strengthen the methodological approach and conclusions.
I have a few minor comments, that are suggestions. I am happy to leave it up to the authors to decide whether they wish to incorporate them.
It is good to go from my perspective. Thank you for an enjoyable review!

*Minor comments*
Line17 / Line 528: The statement that 91% of the world’s surface devoted to livestock production is extensive grazing land is fascinating but has no reference. I appreciate that the abstract likely does not allow references, but I for one would be very interested to know where this figure comes from. Adding the reference to line 528 would be useful I think for many readers.
Figure 1: This is improved and helps create the associations between the methods and factors and activites being observed and managed. I didn’t see any reference to section 1A in the text though? Line 51 refers to Figure 1, and lines 92 and 94 to Figure 1B. This is an extremely minor point, but having taken the time to produce 1A, it may be worth referring to it where you discuss the tools used to document behaviours (lines 52 – 54).
Line 54: The phrase “PLF consists in….” would read better if it were changed to “PLF utilizes….” or similar
Line 439: “a” is unnecessary
Line 524: Please be (briefly) explicit about why more environmentally and wildlife friendly livestock management practices have economic benefits – is it because they can charge more for their products to environmentally conscious consumers? Or because the obtain some value from wildlife such as through tourism? Or some other process?

---

## Round 0.3 · accepted · Accept

Thank you for making the final changes. No further revision is required.

#